# Validation of the Therapeutic Self-Care Scale-European Portuguese Version in Primary Care Type 2 Diabetes Adults

**DOI:** 10.3390/ijerph19073750

**Published:** 2022-03-22

**Authors:** Ana Filipa Cardoso, Paulo Queirós, António Salgueiro Amaral, Carlos Fontes-Ribeiro, Amorim Rosa, Rui Cruz, Matilde Agostinho Neto, Helena Felizardo, Souraya Sidani

**Affiliations:** 1Health Sciences Research Unit, Nursing, Nursing School of Coimbra, 3004-011 Coimbra, Portugal; pauloqueiros@esenfc.pt (P.Q.); amaral@esenfc.pt (A.S.A.); amorim@esenfc.pt (A.R.); helenaf@esenfc.pt (H.F.); 2Faculty of Medicine, University of Coimbra, 3000-370 Coimbra, Portugal; cribeiro@fmed.uc.pt; 3Department of Pharmacy, ESTESC-Coimbra Health School, Polytechnic Institute of Coimbra, 3046-854 Coimbra, Portugal; ruic@estescoimbra.pt; 4Centre for Health Studies & Research, University of Coimbra, 3004-512 Coimbra, Portugal; 5Family Medicine Intern, A Unidade de Saúde Familiar (USF) Casa dos Pescadores, AceS Póvoa de Varzim—Vila do Conde, 4480-807 Vila do Conde, Portugal; amasantos@arsnorte.min-saude.pt; 6School of Nursing, Ryerson University, Toronto, ON M5B 2K3, Canada; ssidani@ryerson.ca

**Keywords:** therapeutic self-care scale, self-care, psychometric properties, type 2 diabetes mellitus

## Abstract

Self-care is an important nursing-sensitive outcome. Reliable and valid measures are needed for therapeutic self-care assessment that may inform the development and evaluation of individualized nursing interventions co-created with type 2 diabetes mellitus (T2DM) adults. The therapeutic self-care scale European Portuguese version (TSCS-EPV) is a validated generic measure that may be used to assess self-care in T2DM adults. Aim: To examine the psychometric properties of the TSCS-EP version in T2DM adults, in primary health care. Methods: A cross-sectional pilot study in a convenience sample of 80 adults with T2DM from two primary health care centers in Portugal was conducted. Individuals completed the Portuguese version of the TSC scale. Results: A three-factor solution emerged from the principal component analysis: “Recognizing and managing signs and symptoms”; “Managing changes in health condition” and “Managing medication”, explaining 75% of the total variance. Total scale Cronbach’s alpha was 0.884 and for the three factors ranged from 0.808 to 0.954. Conclusion: the therapeutic self-care scale European Portuguese version is a promising scale for assessing therapeutic self-care abilities in adults with T2DM in primary care settings. More consistent results on its validity and reliability are needed for it to be used in the country.

## 1. Introduction

The prevalence of Diabetes Mellitus (DM) in the adult population has reached alarming proportions. The global diabetes prevalence in 2019 is estimated to be 9.3% (463 million people), rising to 10.2% (578 million) by 2030 and 10.9% (700 million) by 2045 [1]. Over 60 million people in the European region have DM, and it is projected that by 2045, there will be 68 million people with DM [2]. Portugal has one of the highest prevalence rates (14.2% in 2020) in adults aged 20 to 79 years [1].

Type 2 diabetes (T2D) is the most common type of diabetes, accounting for around 90% of all diabetes worldwide [1]. T2D represents a substantial economic burden to individuals, families and countries’ health systems. However, a significant proportion of the human and economic impact of diabetes is potentially avoidable [3]. Consequently, cost-effective and efficient strategies are needed to assist people in managing Type 2 diabetes mellitus (T2DM) [4] and to secure quality of care and quality of life for people living with diabetes [1]. One such strategy is improving patients’ self-care ability and behaviors as it is essential to successfully achieve positive health outcomes [5,6] and reduce related costs [4]. 

Therapeutic self-care (TSC) is a core patient-oriented outcome for nursing care [7]. Sidani and Doran (2014) defined therapeutic self-care as the perceived ability to engage in self-care behaviors or activities targeting the management of illness. In chronic illness such as T2DM, it is viewed as a decision-making process that focuses on the perceived ability to engage in self-care behaviors or activities targeting the management of illness. Adults with T2DM are expected to develop the ability to successfully manage the medical, behavioral and emotional demands of their illness [8], which involve: (1) recognizing and effectively managing changes in health condition (i.e., symptoms); (2) identifying, selecting and implementing relevant strategies or activities to successfully and promptly manage the changes; and (3) carrying out the recommended treatment regimen (i.e., taking medications as prescribed, performing regular activities of daily living) as well as (4) making decisions about maintaining health and preventing complications and seeking professional care or handling a problem on their own [7,9,10,11]. 

TSC requires a high level of self-care ability since patients only see health professionals a few times a year [4]. From a nursing perspective promoting therapeutic self-care is essential. Accurate assessment of self-care ability is a prerequisite to: understanding individuals’ needs for care and resources [12]; patients’ readiness to engage appropriately in self-care activities and assume the responsibility related to decision-making and implementation of interventions at home [7]; empowering patients with the knowledge and skills they need to appropriately implement strategies addressing self-care demands; achieving positive health outcomes, such as quality of life and well-being; and decreasing the number of hospital admissions, healthcare costs and deaths [6,7,11].

Doran, Sidani, Keatings and Doidge (2002) developed the therapeutic self-care scale (TSCS) to assess the level of self-care ability as perceived by patients. It is composed of 12 items focused on knowledge or ability to engage in activities that patients can perform to meet their health care needs [13]; the activities include taking medications as prescribed, recognizing and managing symptoms, following prescribed treatments, accessing appropriate resources for medical emergencies, performing activities of daily living and managing changes in health condition [13,14].

The TSCS can be administered by nurses during their encounter with patients or significant others, or after providing patient education aimed at promoting self-care; in the latter case, patients’ responses may reflect their understanding of what was taught. To date, only a few studies have validated the TSCS, and these were conducted in acute care settings [12,13,15,16].

Therapeutic self-care in adults with T2DM has been usually assessed by disease-specific instruments, which cover self-care activities that patients with a particular disease are expected to perform. However, the use of these instruments constrains the opportunity to compare self-care in which the patient’s different chronic diseases engage. 

The lack of valid and reliable generic self-care measures may account for the limited assessment of this nursing-sensitive outcome in clinical practice, particularly in Portugal. Given that the TSCS assesses patients’ self-reported ability to care for themselves and manage their illness demands or needs, which is an important prerequisite for T2DM care management, it can be a useful tool to assist primary health care nurses to identify the learning needs for health management of patients with T2DM. However, the TSCS has not been tested for its reliability and validity in this patient population. 

Given the lack of a valid and reliable generic measure in Portugal for assessing self-care ability in adults with T2DM, this study aimed to examine the psychometric properties of the European Portuguese version of the therapeutic self-care scale in adults with T2DM in primary health care settings.

## 2. Materials and Methods

### 2.1. Participants

A cross-sectional design was used in this methodological study. A consecutive sampling strategy was used. The total sample consisted of 80 adults with T2DM, which is adequate considering the criteria set by Bryman and Cramer (2005) [17] of having no less than 50 participants and a ratio of 3 to 5 participants per item. Inclusion criteria were: having at least 18 years of age, having a T2DM diagnosis, being able to read and write in Portuguese, being cognitively intact, and accepting to participate in the study on a voluntary basis. The exclusion criteria were: having a psychiatric illness history and being unable to answer the questions.

### 2.2. Instruments and Procedures

The study took place in two health care centers in the central region of Portugal. Active strategies were used to recruit T2DM adults: the researcher informed healthcare professionals of the study aims and asked them to invite eligible adults with T2DM for the study. The main researcher presented the study to the adults with T2DM while they were waiting for their medical or nursing appointment and administered the data collection instruments in an interview format to those who accepted to participate.

### 2.3. Variables and Measures

Therapeutic self-care scale: The therapeutic self-care scale European Portuguese version was administered to measure patients’ perception of self-care ability. The TSCS-EP version was translated and adapted to European Portuguese by Cardoso, Queirós, Ribeiro and Amaral (2014) [18] according to recommended international guidelines. The scale has shown reliability (Cronbach’s alpha = 0.97) and validity (one factor explaining 81, 3% of the total variance) [18]. This scale included 12 items that assess knowledge about or the ability to engage in relevant behaviors related to taking medications as prescribed by the doctor, recognizing and managing symptoms, following prescribed treatments, accessing appropriate resources for medical emergencies, performing activities of daily living and managing changes in health condition. For the TSCS-EP version, the items are scored on a numeric rating scale, anchored with not at all (0) and very much so (5). High scores indicate high levels of self-care ability [13]. For this study, to improve patients understanding of items 4, 5, 6 and 7, it was necessary to introduce the same example. T2DM participants were asked to (i) identify and describe the signs and symptoms of hypoglycemia and hyperglycemia, as well as (ii) demonstrate their knowledge about the need to have a source of fast-acting carbohydrates with them as a safety measure for hypoglycemia.

Socio-demographic and clinical characteristics: Participants were asked to provide information on their socio-demographic characteristics, including age, gender, marital status, and level of education. Clinical characteristics encompassed: length of diagnosis, engagement in physical activity, medication intake, hospital admissions in the past year due to complications, comorbidities (hypertension, microalbuminuria, renal insufficiency, stroke, obesity, retinopathy, diabetic foot, ischemic coronary heart disease, other). All participants were screened for functional status based on the Lawton and Brody Scale [19] and health locus of control using the health locus of control instrument developed by Ribeiro (1994) [20]. One closed question was used to assess exposure to formal education on T2DM. 

### 2.4. Statistical Analyses

Statistical analysis was performed using SPSS for Windows. (Statistical Package for the Social Sciences version 24, IBM Corp., Armonk, NY, USA). Descriptive statistics (frequency distribution and measures of central tendency and dispersion) were used to describe participants’ socio-demographic profile, clinical characteristics and responses to the items comprising the TSCS-EP version. The percentage of participants who completed each item was calculated to assess the acceptability of the items’ content to participants.

The reliability was examined by computing: (1) the Cronbach’s alpha coefficient, with values greater than 0.70 considered acceptable [21]; (2) the item-total correlation coefficients, which should be above 0.30 [22]; and (3) the inter-item correlation coefficients, which should be above 0.30 [22]. 

Construct validity was assessed by exploratory factorial analysis (EFA) using the principal component method and varimax orthogonal rotation [22,23]. Principal component and orthogonal rotation are the most used in exploratory factor analysis; the aim is to obtain a smaller number of factors with high loadings, which results in more interpretable factors [22,23].

The sample adequacy criterion was confirmed using the Kaiser–Meyer–Olkin (KMO) suitability test. Sphericity was explored using Bartlett’s test. KMO values from ≥0.7 were considered suitable for factor analysis. Kaiser’s criterion was used to retain factors with an eigenvalue larger than 1.0, after varimax orthogonal rotation [22]. The criteria for extraction and interpretation of factors were: (i) commonalities coefficients greater than 0.45; (ii) items with factor loadings ≥ 0.40; (iii) cross loadings ≥ 0.20 with the remaining factors [22]; and also if there was an agreement between the theoretical structure underlying the instrument and the factor solution found.

### 2.5. Ethical Considerations

The study was approved by the Ethics Committee of the Faculty of Medicine of University of Coimbra, Portugal (decision number 53-CE-2011). All ethical and legal principles were met. The study was conducted in accordance with the Declaration of Helsinki [24]. Participation was voluntary, and information about the purpose and nature of the study was given to all participants that signed an informed consent form.

## 3. Results

### 3.1. Sociodemographic and Clinical Characteristics of Study Participants

The participants were mostly men (60%), with a mean age of 60.9 ± 8.63 years; 56.3% were aged 51 to 65 years. Most participants were married/cohabiting (77.5%), had completed four years of elementary school (43.8%), and were retired (66.3%). The majority (95%) of participants had a family nurse. The mean time since diagnosis was 7.9 ± 7.28 years. The majority of participants reported taking medications for different health conditions (93.7%), and of these, 86.7% took oral medication for T2DM. More than half (58.7%) of the participants indicated they engaged in physical activity. Most participants (75%) reported never having received specific diabetes education. In the year prior to the study, 90% of the participants had not been hospitalized due to diabetes. The most common comorbid diseases were hypertension (65.0%), obesity (31.3%), diabetic foot (12.5%), and retinopathy (10%) (Table 1).

### 3.2. Responses to Therapeutic Self-Care Scale European Portuguese Version

On average the European Portuguese version of the TSCS-EP version took 10 min to complete. Overall, the scale was very well accepted by most participants, evidenced by the 100% completion rate (i.e., no missing data). Item 10, which assesses people’s ability to act in case of medical emergency, had the highest mean score (4.94 + 0.559), whereas item 4, which assesses patients’ ability to recognize changes in their body related with illness or health condition had the lowest (2.39 + 1.739). The mean for the total TSCS-EP version score was 44.50 ± 11.63 (possible range 0 to 60), implying that participants showed a good level of self-care ability (Table 1).

### 3.3. Construct Validity of the Therapeutic Self-Care Scale European Portuguese Version

Kaiser–Meyer–Olkin values (KMO = 0.797) and Bartlett’s test approximated chi-square (745.386) (*p* < 0.001) confirmed the sampling adequacy for performing EFA [21]. A succession of factor analyses yielded a three-factor solution with eigenvalue greater than one, and cumulatively explained 75% of the total variance. Table 2 shows the factor loadings after rotation.

The factors were conceptually relevant, and all 12 items were retained. Commonalities were greater than 0.45 with the exception of item 11, which loaded 0.354, indicating a lower explanatory power of the variable by the factor, but it was kept due to its theoretical relevance to the construct. The items loaded on the respective factors as expected, except for items 8 and 12, which had similar relative loadings on two factors (cross loadings < 0.20). However, the items were kept since they were very relevant for the construct and were allocated to factors that would ensure greater consistency with the conceptual framework that guided the development of the scale. 

The first factor included four items (4, 5, 6, and 7) and explained 30.83% of the variance in the items’ responses, and the factor loadings ranged from 0.854 to 0.917. This first factor corresponded to the domain entitled “Recognizing and managing signs and symptoms”. The second factor included five items (8, 9, 10, 11, and 12) and explained 23.87% of the variance in the items’ responses, with factor loadings ranging from 0.468 to 0.904. This factor corresponded to the domain entitled “Managing changes in health condition”. Finally, the third factor encompassed three items (1, 2, and 3) and explained 20.27% of the total variance, with factor loadings ranging from 0.790 to 0.884. This domain was entitled “Managing medication” (Table 2).

Statistically significant associations were found between the TSCS-EP version total and subscale scores and some conceptually relevant variables. Older people with T2DM reported poorer overall self-care ability (r_s_ = −0.26; *p* = 0.018) as well as poorer ability to manage their health condition (r_s_ = −0.23; *p* = 0.041) and ability to recognize and manage signs and symptoms (r_s_ =−0.22; *p* = 0.046). Participants with high education levels reported high overall self-care ability (*p* = 0.025) and ability to recognize and manage signs and symptoms (*p* = 0.017). Individuals with external locus of control reported poorer overall self-care ability (r_s_ = −0.34; *p* = 0.002), abilities to manage medication (r_s_ = −0.32; *p* = 0.004), and abilities to recognize and manage signs and symptoms (r_s_ = −0.27; *p* = 0.016).

### 3.4. Reliability of the Therapeutic Self-Care Scale European Portuguese Version

The reliability of the total TSCS-EP version was evidenced with item–total correlations ranging between 0.424 and 0.925, and a Cronbach’s alpha value of 0.884 [21,22,23]. Cronbach’s alpha values for the three factors ranged from 0.808 to 0.954 (Table 3), which indicate a good/excellent internal consistency [21,22,23]. The results also showed that the reliability of the factor “Recognizing and managing signs and symptoms” is higher than the overall scale reliability, which may reflect high variability in this sample’s responses to the items. Although the reliability of the factor “Managing medication” could be improved if item 3 is omitted, the item was kept because it is a relevant empirical and clinical indicator and has a theoretical relevance for the construct.

## 4. Discussion

Effective therapeutic self-care is a key health outcome for adults with T2DM. The assessment of T2DM patients’ ability to engage in self-care activities is essential as it provides relevant information for primary care nurses or other health care professionals to individualize interventions. The TSCS-EP version was initially developed to measure self-care ability as perceived by patients with acute health conditions requiring medical and/or surgical treatment [14]. This study’s results support its potential utility in assessing the self-care ability of patients with T2DM in primary care. The scale was very well accepted by the participants as evidenced by the 100% completion rate. 

The results of the exploratory factor analysis supported a three-factor structure of the TSCS-EP version. The designation of these factors was based on the conceptual framework underpinning the TSCS and the findings of previous studies [12,14,25]. The three factors represent three core domains of self-care expected of patients with T2DM and include the process of: “Recognizing and managing signs and symptoms” (items 4, 5, 6 and 7); “Managing changes in health condition” (items 8, 9, 10, 11 and 12) and the selection and implementation of appropriate strategies when disease exacerbation is detected; and effectively “Managing medication” (items 1, 2, and 3) [26].

The three emerging factors are in line with those found by Chaboyer, Ringdal, Aitken, and Kendall (2013) who accepted a three-factor solution (“Recognizing and managing signs and symptoms”; “Managing changes in health condition” and “Managing medication”) after deleting items 8 and 11. Sidani (2008) also found a four-factor solution and Richard (2014) found a forced two-factor solution (“Self-management” and “Self-care”). 

In the previous adaptation and validation study performed by Cardoso, Queirós, Ribeiro, and Amaral (2014), a single factor was found for patients with acute diseases. The different factor structure that emerged in this study may indicate that people with acute illness perceived self-care ability differently from those with chronic conditions.

Self-care levels in people with acute illness seem to be inherently vulnerable to ceiling effects. This may be due to some factors: (i) it is likely that at discharge from hospital, patients may not be entirely aware of the difficulties that self-management at home may require and their needs remain unpredictable [12]; (ii) persons recovering from an acute illness may feel that they will have the right support from their social network (family, friends, professionals) [27]; (iii) they may have benefitted from self-care education provided before discharge, which may temporarily enhance their perception of being able to perform self-care activities [28]; and (iv) individuals may have not been effectively engaged in self-care during hospitalization or even had no interest in being actively involved [29]. 

The analysis of the mean item scores showed that participants in this study reported a moderate-to-high ability to manage changes in health condition and use healthcare resources, which included the performance activities of daily living. It seems that patients with T2DM develop the ability to manage changes in health condition efficiently, possibly as a result of the disease trajectory, or of their regular contact with health professionals, namely family nurses. It is possible that in these contacts, nurses discuss information related to self-care management. On the other hand, adults with T2DM may be more likely to engage in self-care activities throughout their lives because they feel supported by their social network [27]; or have higher readiness to engage in self-care activities that do not limit or affect their social participation.

Adults with T2DM reported lower average scores in medication management when compared to managing changes in health condition and using healthcare resources, but they were still high. Medication management seems to be a highly valued dimension by adults with T2DM. This can be explained by different factors: (i) the time of living with the disease (8 years) that may have contributed to the development of a deeper knowledge about medications; (ii) patients may value medication more than other treatment approaches such as diet or physical activity; (iii) patients may benefit from regular contact with health professionals who help them to manage medication; or even (iv) these results may be explained by the external locus of control, suggesting that patients with T2DM understand that the treatment of the disease depends more on aspects external to their responsibility and direct participation.

Adults with T2DM also perceived a low ability in recognizing and managing signs and symptoms. Considering that T2DM is mostly asymptomatic, it may explain why patients may not feel able or confident to recognize signs and symptoms. In general, participants did not experience any changes related to the disease (hyper/hypoglycemia), which may be associated with the adequacy of the therapeutic control. Results also highlighted that 75% of the individuals had not received formal education on T2DM. This finding highlights the importance of providing formal education and support to promote successful self-care management of T2DM.

Nurses can support and empower adults with T2DM to develop the knowledge and skills to effectively manage their medication regimen; to recognize and manage signs and symptoms and also to manage changes in health condition and using healthcare resources, specifically in those with less developed skills or lower levels of self-care autonomy. Evidence based person-centered interventions such as motivational interviewing, technology-based interventions, education, lifestyle modification programs, mindfulness, coaching/peer health coaching/peer support may be used to empower people to develop therapeutic self-care ability [30].

The overall TSCS-EP version and the three factors demonstrated good reliability (Cronbach’s alpha > 0.80) [21,22,23]. This result is consistent with those found by Sidani (2008) and Chaboyer, Ringdal, Aitken, and Kendall (2013). The high item-total and inter-item correlation also indicates a good correlation among all items. Cronbach’s alpha values did not decrease when items were deleted, overall. It also shows that the TSCS-EP version items are suitable to assess self-care ability in adults with T2DM. The high consistency in the presence of multidimensionality indicates that the items that compose the different factors are strongly correlated, and that the factors operationalize different domains of the same concept, that is, self-care ability. Further, the items are homogeneous, and the homogeneity of the items of one or more dimensions may be higher than that obtained when considering the totality of the items, as noticed in the results of this study [23]. Participants who took no medication reported having difficulties in answering items 1–3 about managing medication. The lack of an adequate response option may have conditioned some answers; hence, a response option such as “not applicable” would be useful for these items.

### Strengths and Limitation of the Study

This study’s major strength is that it determined the validity and reliability of this TSCS-EP version. The cross-cultural adaptation process followed rigorous quality procedures, resulting in the first version of an instrument for assessing self-care ability in primary health care settings. A potential limitation of this study is the sample size. Further studies should be conducted using larger samples and with different characteristics to reinforce the validity and reliability in adults with T2DM or even in other chronic conditions, as well as to explore its association with other variables.

## 5. Conclusions

Therapeutic self care is a nursing-sensitive outcome, and its measurement allows the assessment of patients’ needs and the delivery of individualized care. It is particularly important to assess the self-care ability of patients with T2DM. This study was conducted due to the lack of a measurement instrument to assess therapeutic self care in adults with T2DM in the Portuguese context. The TSCS-EP version has shown great reliability, validity and acceptability in this sample of adults with T2DM. Three dimensions have emerged: “Recognizing and managing signs and symptoms”, “Managing changes in health condition” and “Managing medication”. Therefore, it contributes to the conceptualization of TSC domains in adults with T2DM and the identification of areas with insufficient knowledge, skills, and resources. It is a simple, easy, and friendly instrument and can be relevant for primary care research and practice.

## Figures and Tables

**Table 1 ijerph-19-03750-t001:** Descriptive statistics for the Therapeutic Self-Care Scale European Portuguese version items.

	TSCS-EP Version Items *	Mean	Standard Deviation	N
1	Do you know what medications you have to take?	3.73	1.669	80
2	Do you understand the purpose of the medications prescribed to you (that is, do you know what the medications do for your health condition)?	3.61	1.563	80
3	Do you take the medications as prescribed?	3.85	1.519	80
4	Can you recognize changes in your body (symptoms) that are related to your illness or health condition?	2.39	1.739	80
5	Do you know and understand why you experience some changes in your body (symptoms) related to your illness or health condition?	2.44	1.749	80
6	Do you know what to do (things or activities) to control these changes in your body (symptoms)?	2.55	1.820	80
7	Do you carry out the treatments or activities that you have been taught to manage these changes in your body (symptoms)?	2.75	1.859	80
8	Do you do things or activities to look after yourself and to maintain your health in general?	4.25	1.373	80
9	Do you know whom to contact to get help in carrying out your daily activities?	4.86	0.689	80
10	Do you know whom to contact in case of a medical emergency?	4.94	0.559	80
11	Do you perform your regular activities (such as bathing, shopping, preparing meals, visiting with friends)?	4.86	0.725	80
12	Do you adjust your regular activities when you experience body changes (symptoms) related to your illness or health condition?	4.28	1.441	80

Note: * Items are presented as in the original scale. If there is any interest in the European Portuguese scale, please contact the authors.

**Table 2 ijerph-19-03750-t002:** Principal components matrix after orthogonal varimax rotation of items and commonalities.

Rotation Component Matrix				
	Components			
TSCS-EP Version Items	1	2	3	H^2^
4	**0.854**	0.021	0.272	0.830
5	**0.891**	0.059	0.273	0.861
6	**0.913**	0.106	0.254	0.690
7	**0.917**	0.145	0.148	0.803
8	0.352	**0.468**	0.104	0.871
9	−0.028	**0.827**	0.200	0.910
10	0.020	**0.904**	0.149	0.884
11	0.063	**0.869**	0.084	0.354
12	0.440	**0.513**	0.083	0.725
1	0.269	0.225	**0.841**	0.839
2	0.241	0.149	**0.884**	0.766
3	0.222	0.131	**0.790**	0.463
Eigenvalues	5.359	2.322	1.3217	
% of variance	30.835	23.874	20.276	

Bold: To Highlight the factor loading.

**Table 3 ijerph-19-03750-t003:** Therapeutic Self-Care Scale European Portuguese version dimension reliability.

TSCS-EP Version Subscale	Items	$\bar{x}$	SD	Corrected Item-Total Correlation	Cronbach’s Alpha If Item Deleted
Recognizing and managing signs and symptoms α = 0.954	4	2.39	1.739	0.847	0.950
	5	2.44	1.749	0.901	0.935
	6	2.55	1.820	0.925	0.927
	7	2.75	1.859	0.875	0.943
Managing changes in health condition α = 0.808	8	4.25	1.373	0.424	0.715
	9	4.86	0.689	0.552	0.661
	10	4.94	0.559	0.663	0.652
	11	4.86	0.725	0.650	0.628
	12	4.28	1.441	0.466	0.703
Managing medication α = 0.871	1	3.73	1.669	0.814	0.762
	2	3.61	1.563	0.824	0.754
	3	3.85	1.519	0.635	0.919
Global internal consistency α = 0.884					

## Data Availability

The data presented in this study are available on request from the corresponding author. The data are not publicly available because this issue was not considered within the informed consent signed by the participants of the study.

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
