# Peer review of "Validation of the Therapeutic Self-Care Scale-European Portuguese Version in Primary Care Type 2 Diabetes Adults"

_ijerph, 2022, doi:10.3390/ijerph19073750_

Round 1
Reviewer 1 Report
Please, authors should provide an explanation or justification for the following issues:
- Explain or justify the using of Varimax orthogonal rotation (page 4, paragraph 4).
- Why do authors propose items with factor loadings ≥ 0.30? (page 4, paragraph 4). So long ago, the minimum accepted are factor loadings of 0.40.
- Please delete Table 1, it is irrelevant since that information is already shown in Table 3.
- Speaking of Cronbach´s alpha values, I suggest the use of three decimals, even in the text.
- In Table 2, please show all the factor loadings values for all the components. Item 12 should be deleted from component 2. If authors would adjust to the 0.40 cut-off on factor loadings, there would be no objection to item 8.
- At the title of the article, I suggest mentioning something related to "primary care DM2 patients".
- It is not clear why the authors´ analysis ended with specific Cronbach´s alpha values superior to the global internal consistency (α = 0.884). Please justify and explain.
Author Response
Validation of the Therapeutic Self-Care Scale - European Portuguese version in primary care type 2 diabetes adults
We would like to thank for all the comments and suggestions given by the reviewers that help us to improve our manuscript.
Please, see below how the reviewer’s comments have been addressed.
To facilitate reviewer’s analysis, comments were added in the text and highlighted with red color.
#Reviewer 1
Comment 1: Explain or justify the using of Varimax orthogonal rotation (page 4, paragraph 4)
Response: More information was provided in line 162. The following sentence was added: “The varimax method is the most used in exploratory factor analysis. Furthermore, this method was used to obtain a smaller number of variables with high loadings on each factor, which results in more interpretable factors”
Comment 2: Why do authors propose items with factor loadings ≥ 0.30? (page 4, paragraph 4). So long ago, the minimum accepted are factor loadings of 0.40.
Response: The information about the factor loadings was incorrect. The information was corrected to 0,40, according to Field et al. and following the suggestion.
Comment 3: Please delete Table 1, it is irrelevant since that information is already shown in Table 3.
Response: We kindly apologize, but the comment wasn’t clear for us. In table 1 we present information about mean and standard deviation that support previous data e that is not stated on table 3. We introduced the original item designation to improve understanding.
Comment 4: Speaking of Cronbach´s alpha values, I suggest the use of three decimals, even in the text.
Response: Three decimals have been inserted and corrected in the text.
Comment 5. In Table 2, please show all the factor loadings values for all the components. Item 12 should be deleted from component 2. If authors would adjust to the 0.40 cut-off on factor loadings, there would be no objection to item 8
Response: Information about all factors loadings was added;
Comment 6. In Table 2, please show all the factor loadings values for all the components. Item 12 should be deleted from component 2. If authors would adjust to the 0.40 cut-off on factor loadings, there would be no objection to item 8
Response: We understand that the item 12 has a cross loading inferior to 0,20, however it is theoretically relevant to the construct and more coherent with the factor 2 and that is justified in the text. Cut-off of factor loadings was updated to 0.40.
Comment 7. At the title of the article, I suggest mentioning something related to "primary care DM2 patients".
Response: The suggestion was considered and the title was changed to “Validation of the Therapeutic Self-Care Scale - European Portuguese version in primary care type 2 diabetes adults”
Comment 8. It is not clear why the authors´ analysis ended with specific Cronbach´s alpha values superior to the global internal consistency (α = 0.884). Please justify and explain.
The following sentence was introduced (line 258):
“The results also shown that the reliability of the dimension “Recognizing and managing signs and symptoms” is higher than overall reliability. Even though the dimension’s reliability was high, concerning “Managing medication” dimension reliability it resulted from our analysis that if …”
The following sentence was introduced (line 348):
“The high consistency in the presence of multidimensionality indicates that the items that compose the different dimensions of this scale are strongly correlated. In other words, it means that the items are homogeneous and the homogeneity of the items of one or more dimensions may be higher than that obtained when considering the totality of the items, as noticed in the results of this study.”
Other minor changes |
||
Line |
Changes type |
Information added/changed |
16 |
The information was updated |
Polytechnic Institute of Coimbra, ESTESC-Coimbra Health School, Pharmacy, 3046-854 Coimbra, Portugal. Centre for Health Studies & Research, University of Coimbra, Coimbra, Portugal |
219 |
The following sentence was added |
Table 2 shows the factor loadings after rotation. |
246 |
The sentence was deleted because the information didn’t add anything to the paper. |
“Those with higher perception of nursing professional support for decision-making had better self-care ability for managing their health condition. (rs=0.25; p=0.026).” |
255 |
The following sentence was improved |
“an overall reliability of the questionnaire was good with a Cronbach's alpha value of 0.884” |
256 |
The following sentence was improved |
Cronbach's alpha values for the three dimensions ranged from 0.808 to 0.954 (Table 3) which indicate a good/excellent internal consistency [22, 23, 24]. |
309 |
The previous sentence was changed into the following |
“or even patients may benefit from regular contact with health professionals, namely family nurses.” |
314 |
The following sentence was improved |
“Adults with T2DM reported lower averages scores in medication management domain when compared to managing changes in health condition and using healthcare resources, but still high. Medication management seems to be highly value dimension by T2DM adults.” |
330 |
The following sentence was improved |
Results also highlighted that 75% of the individuals had not received formal education on T2DM which underscore the importance to provide formal education and support needed to promote successfully self-care management of T2DM |
343 |
References were added and the sentence was improved
|
“The overall TSCS-EP version and the three dimension’s demonstrated good reliability (Cronbach's alpha >0.80 ) [22, 23, 24]. This result is consistent with those found by Sidani (2008) and Chaboyer, Ringdal, Aitken, and Kendall (2013). The high item-total and inter-item correlation also indicates a good correlation among all items.” |
380 |
The following sentence was added |
Funding: This work is funded by national funds through FCT—Portuguese Foundation for Science and Technology, I.P., within the scope of project Ref. UIDB/00742/2020 . |
391 |
The following sentence was added |
Acknowledgments: The authors gratefully thank to all the participants of this study. The authors also thank the support of the Health Sciences Research Unit: Nursing, Nursing School of Coimbra, Portugal . |
462 |
The reference was change (the previous was wrong) |
23. Field A, Miles J, Field Z. Discovering statistics using R. Thousand Oaks: Sage; 2022. 992 p . |
463 |
The reference was added |
24. Maroco, João, and Teresa Garcia-Marques. "Qual a fiabilidade do alfa de Cronbach? Questões antigas e soluções modernas?." [Internet]. Laboratório de psicologia 4.1 2006: 65-90 . |

Reviewer 2 Report
I commend you for not only doing the usual reliability analyses. Using principal component analysis gives an objective view of the data.
Also, I commend you for the explanation of the sample and item ratio.
Author Response
Validation of the Therapeutic Self-Care Scale - European Portuguese version in type 2 diabetes adults
We would like to thank for all the comments and suggestions given by the reviewer that help us to improve our manuscript.
# Reviewer 2
I commend you for not only doing the usual reliability analyses. Using principal component analysis gives an objective view of the data. Also, I commend you for the explanation of the sample and item ratio.
We would like to thank for the comments.

Reviewer 3 Report
- Line 128-129: The scale has shown reliability and validity. Provide reference.
- Line 229-232: Older people with T2DM reported poorer scores. It is assumed that their mental health has no effect on the results. However, factors such as overall physical health, complications such as high blood pressure and other age related factors were not considered.
- The effects of self-care education was mentioned but not included in the study.
- Factors that led to higher or lower scores would have been more beneficial to highlight in the study rather that jus the scores.
Author Response
Validation of the Therapeutic Self-Care Scale - European Portuguese version in primary care type 2 diabetes adults
We would like to thank for all the comments and suggestions given by the reviewers that help us to improve our manuscript.
Please, see below how the reviewer’s comments have been addressed.
To facilitate reviewer’s analysis, comments were added in the text and highlighted with red color.
Reviewer 3
Comment 1. Line 128-129: The scale has shown reliability and validity. Provide reference.
Response: The reference about the primary reference was changed to the bottom of the sentence to improve understanding
Comment 2. Line 229-232: Older people with T2DM reported poorer scores. It is assumed that their mental health has no effect on the results. However, factors such as overall physical health, complications such as high blood pressure and other age related factors were not considered.
Response: We kindly apologize, but the comment wasn’t clear for us. The reviewer affirmed that we assumed that their mental health has no effect on the results. In this study we didn’t assess any health mental outcome. Information about lower self-care older people with T2DM was provided.
Comment 3. The effects of self-care education was mentioned but not included in the study.
Response: We kindly apologize but the question raised by the reviewer wasn’t clear for us. It wasn’t clear what effects of self-care education do the reviewer is referring to. If needed, we kindly ask to provide further information on this comment.
In line 191 information about level of education academic was provided;
In line 195 information about formal education on diabetes was provided;
In line 238 information about Statistically significant associations that were found is provided. However, no statistical association was found between diabetes formal education and self-care.
Comment 4. Factors that led to higher or lower scores would have been more beneficial to highlight in the study rather that jus the scores
Response: Once this study aimed to determine the psychometric properties we decided not to present some factors and will be presented in a future empirical paper.
Other minor changes |
||
Line |
Changes type |
Information added/changed |
16 |
The information was updated |
Polytechnic Institute of Coimbra, ESTESC-Coimbra Health School, Pharmacy, 3046-854 Coimbra, Portugal. Centre for Health Studies & Research, University of Coimbra, Coimbra, Portugal |
219 |
The following sentence was added |
Table 2 shows the factor loadings after rotation. |
246 |
The sentence was deleted because the information didn’t add anything to the paper. |
“Those with higher perception of nursing professional support for decision-making had better self-care ability for managing their health condition. (rs=0.25; p=0.026).” |
255 |
The following sentence was improved |
“an overall reliability of the questionnaire was good with a Cronbach's alpha value of 0.884” |
256 |
The following sentence was improved |
Cronbach's alpha values for the three dimensions ranged from 0.808 to 0.954 (Table 3) which indicate a good/excellent internal consistency [22, 23, 24]. |
309 |
The previous sentence was changed into the following |
“or even patients may benefit from regular contact with health professionals, namely family nurses.” |
314 |
The following sentence was improved |
“Adults with T2DM reported lower averages scores in medication management domain when compared to managing changes in health condition and using healthcare resources, but still high. Medication management seems to be highly value dimension by T2DM adults.” |
330 |
The following sentence was improved |
Results also highlighted that 75% of the individuals had not received formal education on T2DM which underscore the importance to provide formal education and support needed to promote successfully self-care management of T2DM |
343 |
References were added and the sentence was improved
|
“ The overall TSCS-EP version and the three dimension’s demonstrated good reliability (Cronbach's alpha >0.80 ) [22, 23, 24]. This result is consistent with those found by Sidani (2008) and Chaboyer, Ringdal, Aitken, and Kendall (2013). The high item-total and inter-item correlation also indicates a good correlation among all items.” |
380 |
The following sentence was added |
Funding: This work is funded by national funds through FCT—Portuguese Foundation for Science and Technology, I.P., within the scope of project Ref. UIDB/00742/2020 . |
391 |
The following sentence was added |
Acknowledgments: The authors gratefully thank to all the participants of this study. The authors also thank the support of the Health Sciences Research Unit: Nursing, Nursing School of Coimbra, Portugal . |
462 |
The reference was change (the previous was wrong) |
23. Field A, Miles J, Field Z. Discovering statistics using R. Thousand Oaks: Sage; 2022. 992 p . |
463 |
The reference was added |
24. Maroco, João, and Teresa Garcia-Marques. "Qual a fiabilidade do alfa de Cronbach? Questões antigas e soluções modernas?." [Internet]. Laboratório de psicologia 4.1 2006: 65-90 . |

Reviewer 4 Report
The manuscript “Validation of the Therapeutic Self-Care Scale - European Portuguese version in type 2 diabetes adults” by Cardoso et al. develops a scale for self-assessment and knowledge of type 2 diabetes in patients in order to improve the course of the pathology. Although the starting point may be interesting, this work should serve to develop the conclusions that are drawn and finally see if applying these measures improves the quality of life of patients. For that reason I think it is incomplete and should not be published in these circumstances.
On the other hand, the list of questions that are asked to patients does not appear at any time and in this way it is difficult to evaluate the results obtained.
Author Response
Validation of the Therapeutic Self-Care Scale - European Portuguese version in primary care type 2 diabetes adults
We would like to thank for all the comments and suggestions given by the reviewer that help us to improve our manuscript.
Please, see below how the reviewer’s comments have been addressed.
To facilitate reviewer’s analysis, comments were added in the text and highlighted with red color.
Reviewer 4 |
|
The manuscript “Validation of the Therapeutic Self-Care Scale - European Portuguese version in type 2 diabetes adults” by Cardoso et al. develops a scale for self-assessment and knowledge of type 2 diabetes in patients in order to improve the course of the pathology. Although the starting point may be interesting, this work should serve to develop the conclusions that are drawn and finally see if applying these measures improves the quality of life of patients. For that reason I think it is incomplete and should not be published in these circumstances. |
We kindly apologize but the question raised by the reviewer wasn’t clear enough for us. This assessment scale was previous developed by Sidani and Doran and aimed to assessed core self-care therapeutic behaviours. As a methodological study we sought to provide information on scale psychometrics characteristics in T2DM adults. This measure assesses self-care and so it is not possible to assess its effectiveness on quality of life. We believe that the assessment of self-care behaviours through this scale may help to develop interventions that benefit self-care and other results such as quality of life or well-being but it cannot be stated in this study. |
On the other hand, the list of questions that are asked to patients does not appear at any time and in this way it is difficult to evaluate the results obtained. |
We thank for this comment because we also struggled with this question. By one hand, we notice the tendency of this type of studies does not provide the items information in such a way that it is not used by others without proper authorization. By the other hand, as this is part of a cross cultural study, we understood that if the items should be provided, it would make more sense to be in Portuguese, and it would not help in understanding. As so, we decided to include the original items designation. Also a note was added to help future interested |
Other minor changes |
||
Line |
Changes type |
Information added/changed |
16 |
The information was updated |
Polytechnic Institute of Coimbra, ESTESC-Coimbra Health School, Pharmacy, 3046-854 Coimbra, Portugal. Centre for Health Studies & Research, University of Coimbra, Coimbra, Portugal |
219 |
The following sentence was added |
Table 2 shows the factor loadings after rotation. |
246 |
The sentence was deleted because the information didn’t add anything to the paper. |
“Those with higher perception of nursing professional support for decision-making had better self-care ability for managing their health condition (rs=0.25; p=0.026).” |
255 |
The following sentence was improved |
“an overall reliability of the questionnaire was good with a Cronbach's alpha value of 0.884” |
256 |
The following sentence was improved |
Cronbach's alpha values for the three dimensions ranged from 0.808 to 0.954 (Table 3) which indicate a good/excellent internal consistency [22, 23, 24]. |
309 |
The previous sentence was changed into the following |
“or even patients may benefit from regular contact with health professionals, namely family nurses.” |
314 |
The following sentence was improved |
“Adults with T2DM reported lower averages scores in medication management domain when compared to managing changes in health condition and using healthcare resources, but still high. Medication management seems to be highly value dimension by T2DM adults.” |
330 |
The following sentence was improved |
Results also highlighted that 75% of the individuals had not received formal education on T2DM which underscore the importance to provide formal education and support needed to promote successfully self-care management of T2DM |
343 |
References were added and the sentence was improved
|
“The overall TSCS-EP version and the three dimension’s demonstrated good reliability (Cronbach's alpha >0.80 ) [22, 23, 24]. This result is consistent with those found by Sidani (2008) and Chaboyer, Ringdal, Aitken, and Kendall (2013). The high item-total and inter-item correlation also indicates a good correlation among all items.” |
380 |
The following sentence was added |
Funding: This work is funded by national funds through FCT—Portuguese Foundation for Science and Technology, I.P., within the scope of project Ref. UIDB/00742/2020 . |
391 |
The following sentence was added |
Acknowledgments: The authors gratefully thank to all the participants of this study. The authors also thank the support of the Health Sciences Research Unit: Nursing, Nursing School of Coimbra, Portugal . |
462 |
The reference was change (the previous was wrong) |
23. Field A, Miles J, Field Z. Discovering statistics using R. Thousand Oaks: Sage; 2022. 992 p . |
463 |
The reference was added |
24. Maroco, João, and Teresa Garcia-Marques. "Qual a fiabilidade do alfa de Cronbach? Questões antigas e soluções modernas?." [Internet]. Laboratório de psicologia 4.1 2006: 65-90 . |

Round 2
Reviewer 3 Report
The suggested changes and additions have been taken care of.
Author Response
We would like to thank you for all the comments that helped to improve the manuscript.
Reviewer 4 Report
As you well recognize and that is what I am referring to in my answer, I think this methodological study is interesting, but I insist that it would be more interesting to apply the conclusions obtained to practice, which should be the final objective of the work. In this way, it would not remain as a descriptive study, but rather it could be seen how the quality of life of patients improves, focusing on those points where deficiencies have been seen on a day-to-day basis. In this way, it would have a direct application and the study would be more complete and susceptible to publication.
Author Response
Validation of the Therapeutic Self-Care Scale - European Portuguese version in primary care type 2 diabetes adults
Round 2.
Reviewer’s Comment 1.
As you well recognize and that is what I am referring to in my answer, I think this methodological study is interesting, but I insist that it would be more interesting to apply the conclusions obtained to practice, which should be the final objective of the work. In this way, it would not remain as a descriptive study, but rather it could be seen how the quality of life of patients improves, focusing on those points where deficiencies have been seen on a day-to-day basis. In this way, it would have a direct application and the study would be more complete and susceptible to publication.
Response:
We appreciate the reviewer’s comment, and we plan for another study in which we examine the association of self-care behaviors with other health-related patient outcomes. It is important to note that in many health behavior and psychological sciences, it is recommended to validate newly developed or newly translated self-report instruments prior to using them in research and/or practice. This descriptive paper is in alignment with the latter recommendation, and is comparable to others that have been published to report on the validity and reliability of measures. This paper describes the concept being measured at the conceptual and operational levels, which is important for other researchers interested in using the translated measure. The papers also reports the results of factor analysis which suggest that the items are appropriate indicators of the concept; this in turn, is essential in capturing the concept and in accurately interpreting future studies’ findings related to the association of self-care behaviors with health-related outcomes. Please, note that the original scale (in English) has been found to correlate with health outcomes, and is being used in practice.
